# Sentiment Analysis on Streaming User Reviews via Dual-Channel Dynamic Graph Neural Network

**Xin Zhang   Linhai Zhang   Deyu Zhou**[*]
School of Computer Science and Engineering, Key Laboratory of Computer Network
and Information Integration, Ministry of Education, Southeast University, China
{zhangxin, lzhang472, d.zhou}@seu.edu.cn

## Abstract

Sentiment analysis on user reviews has achieved great success thanks to the rapid growth of deep learning techniques. The large number of online streaming reviews also provides the opportunity to model temporal dynamics for users and products on the timeline. However, existing methods model users and products in the real world based on a static assumption and neglect their time-varying characteristics. In this paper, we present DC-DGNN, a dual-channel framework based on a dynamic graph neural network that models temporal user and product dynamics for sentiment analysis. Specifically, a dual-channel text encoder is employed to extract current local and global contexts from review documents for users and products. Moreover, user review streams are integrated into the dynamic graph neural network by treating users and products as nodes and reviews as new edges. Node representations are dynamically updated along with the evolution of the dynamic graph and used for the final prediction. Experimental results on five real-world datasets demonstrate the superiority of the proposed method.

## 1 Introduction

Sentiment analysis on user reviews, inferring the overall sentiment polarity (e.g. 1-5 stars on the review site Amazon) of a user-written review document for a product, has gained popularity with the rapid growth of online review sites such as Amazon, Yelp, and IMDB. Compared to other sentiment analysis tasks (Yang et al., 2019; Zhou et al., 2020, 2021a), not only the text itself but also the user and product information are crucial for the final rating score prediction.

For sentiment analysis on user reviews, early methods incorporate user and product embeddings by training randomly initialized embedding with the classifier (Tang et al., 2015; Chen et al., 2016;

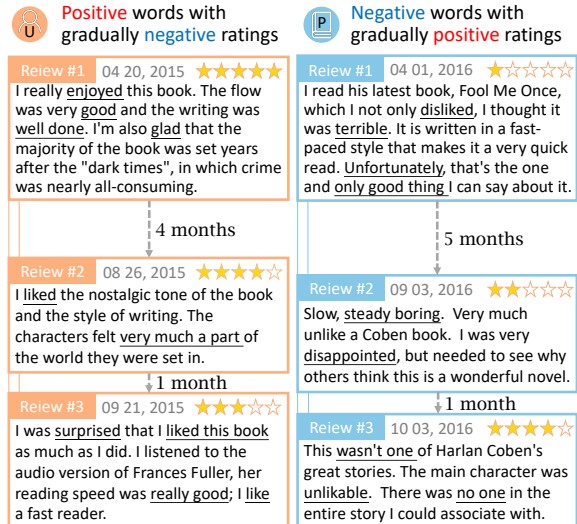

Figure 1: An illustration of Streaming Reviews from a user or to a product. The left part indicates an example of a user who drafts the reviews with the positive words but scores lower with time. The right part indicates an example for a product that receives reviews with negative words but is scored higher with time.

Ma et al., 2017; Dou, 2017). Later, the framework of dual model with text was applied to learn separated representations for user and product (Long et al., 2018; Lyu et al., 2020). Studies have shown that users and products often do not appear as independent individuals and exhibit shared preferences and social interactions (Kim and Srivastava, 2007; Mittal et al., 2022). Therefore, recent attempts improve the embedding quality by implicitly incorporating user-user or user-product relationships (Amplayo et al., 2018; Wen et al., 2020; Zhou et al., 2021c,b).

However, all these aforementioned approaches assume that the user and product characteristics are static, which is unreasonable in the ever-evolving real world. In fact, the users' preferences and the popularity of products are continuously changing over time. Figure 1 illustrates a real case of shifts

---

[*]Corresponding author.

in user rating preference (user $U$) and product rating preference (product $P$) in Amazon Book Review[1]. User $U$ represents an example whose words remain positive while scoring gradually negative. Specifically, in the review on April 20, 2015, some positive words such as *enjoyed*, *well done*, *good* were used with a score of 5 given to the book. In the subsequent two reviews, user $U$ still used positive words, but the ratings were changed to 4 stars and 3 stars. It might indicate that user $U$'s word habits remain consistent, while the scoring preference has shifted, suggesting a more cautious behavior in rating. For product $P$, negative words such as *disliked*, *boring*, and *unlikable* are more likely to appear in the reviews during the display period. However, when analyzing rating changes, product $P$ receives more positive ratings over time, indicating that it has grown in popularity even with negative review documents. Based on the above observations, we assume that dynamic modeling with the temporal user and product dynamics is essential for sentiment analysis on user reviews in real-world changing contexts.

Therefore, in this paper, we propose a new study on sentiment analysis that focuses on streaming user reviews, which takes chronological reviews as a stream and aims to predict the rating score with the dynamic user and product representations. Since modeling pairwise relations between users and products in a graph has been proven useful in sentiment analysis on user reviews (Tan et al., 2011; Zhou et al., 2021c), we provide an intuitive way to model streaming user reviews in a dynamic graph and propose a dual-channel dynamic graph neural network (DC-DGNN) to learn the temporal dynamics of users and products. To interconnect users and products while still preserving their specific properties, DC-DGNN is designed with two dual-channel components based on a first-separate-then-gather strategy. First, a dual-channel text encoder is proposed to encode review documents into the local and global channels to learn separate user-related and product-related contexts respectively. Then, inspired by JODIE (Kumar et al., 2019), a dual-channel dynamic graph updating module is designed to aggregate information of the implicit user-review-product bipartite graph. In this way, dynamic information flows smoothly from one moment to the next with the evolution of the dynamic graph and is integrated into current user and prod-

uct representations. Finally, the updated user and product representations along with the text representation are used for the current time score prediction.

We evaluate the proposed DC-DGNN on five self-built datasets. The experimental results show that DC-DGNN outperforms all existing state-of-the-art methods and the motivation for modeling the temporal dynamics of users and products is verified.

## 2 Related Work

### 2.1 Sentiment Analysis on User Reviews

Incorporating user and product information into models through reviews is the main idea of user review sentiment analysis methods. Tang et al. (2015) shows that modeling the reviewer as well as the product being reviewed is valuable for polarity prediction. Several works (Chen et al., 2016; Ma et al., 2017; Dou, 2017) utilize different semantic-level attentions to encode user and product information. Amplayo (2019) has also looked into the effects of various locations and methods for incorporating attributes into the model. Long et al. (2018) first proposed a dual user and product memory network model to learn their representation separately. Similarly, Lyu et al. (2020) stacks all available historical reviews separately for users and products to enrich their representations.

Recently, to exploit more knowledge from user-user or user-product relations, Amplayo et al. (2018) introduces shared vectors that are constructed from similar users/products to address the cold start problem. Yuan et al. (2019) creates inferred representations from representative users or products to use the inherent correlation between users or products. Wen et al. (2020) aggregates documents written by similar users toward the same product to improve classification accuracy. Zhou et al. (2021b) applies a group-wise model to use group information to supplement user information. However, none of the above works considered the critical factor of temporal information in the real world. In fact, the characteristics of users and products are changing over time. Hence, relying solely on static modeling is not enough to capture their dynamic attributes.

### 2.2 Dynamic Graph Neural Network

Previous graph representation learning mainly focused on static graphs with a predefined number of

---

[1] https://nijianmo.github.io/amazon/index.html

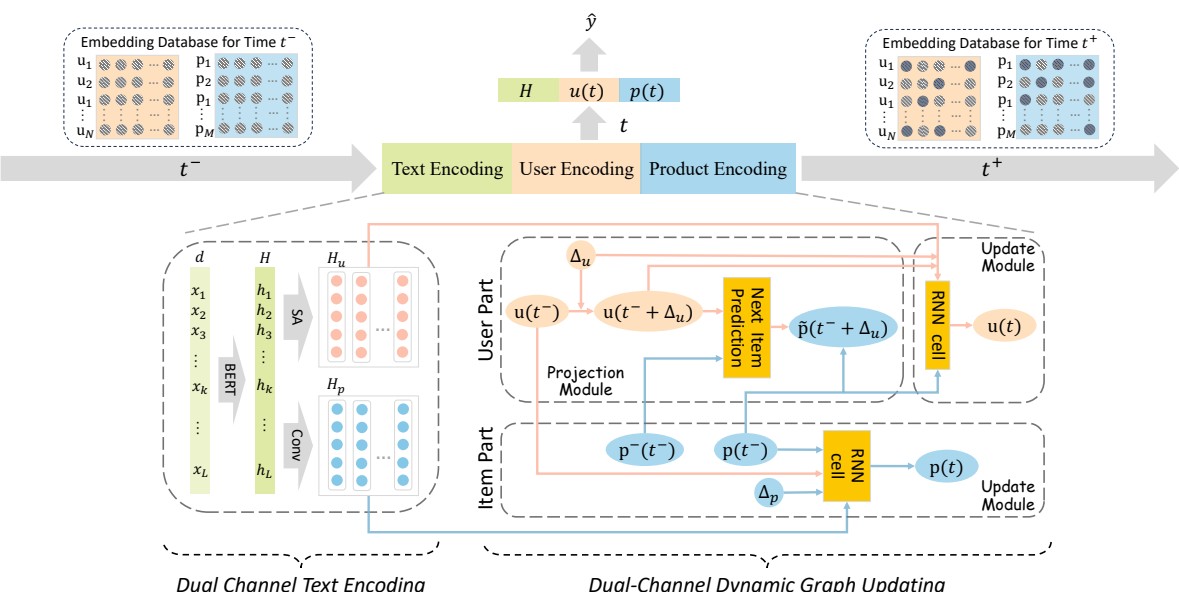

Figure 2: The architecture of Dual-Channel Dynamic Graph Neural Network (DC-DGNN). The dynamic graph evolves through two steps: (1) the review document is sent to *Dual Channel Text Encoding* for Text Encoding, where text representation is encoded into user-related and product-related contexts; (2) the separated contexts are fed into User Part and Product Part in *Dual-Channel Dynamic Graph Updating* for User Encoding and Product Encoding, respectively.

nodes and edges. Real-world graphs, on the other hand, typically change over time as their graph structures change with the nodes and edges coming in or disappearing. In order to tackle real-world situations, Continuous Time Dynamic Graph Neural Network (Nguyen et al., 2018; Kumar et al., 2019) and Discrete Time Dynamic Graph Neural Network (Sankar et al., 2020; Yang et al., 2021) for time dynamic graph representation learning have been proposed. However, despite its potential in modeling real-world graph integration and dynamics, previous studies only focus on its application in the fields of social networks, citation graphs, traffic networks, etc., while research on sentiment analysis still remains blank.

## 3 Methodology

In sentiment analysis on streaming user reviews, reviews are sorted in chronological order in the form of $E = \{\mathcal{E}_1, \ldots, \mathcal{E}_T\}$. Each of them is denoted as $\mathcal{E}_i = (u_i, p_i, t_i, d_i)$, where $t_i$ is timestamp for the review $d_i$, $u_i$ is the user who wrote the review $d_i$ and $p_i$ is the product being reviewed. The task aims to predict the user's rating $y$ towards the product under current condition $\mathcal{E}_t$ based on the historical information $\{\mathcal{E}_1, \ldots, \mathcal{E}_{t-1}\}$, and learns a mapping function between user's rating $y$ and condition $\mathcal{E}_t$, denoted as $y = f(\mathcal{E}_t | \{\mathcal{E}_1, \ldots, \mathcal{E}_{t-1}\})$. Table 1 lists

the important mathematical notations used throughout the paper.

We propose a Dual-Channel Dynamic Graph Neural Network (DC-DGNN) to model the dynamics of the streaming review data, as shown in Figure 2, which takes a sequence of reviews as input and integrates the review information into users and products through two dual-channel components. More concretely, for a document written by a user, a dual-channel text encoding component is first applied to encode text into user-related and product-related channels. In addition, a dual channel dynamic graph updating component is proposed to integrate new review information into users and products. Finally, by concatenating the original text representation, the updated user representation as well as the updated product representation and feeding them into a prediction layer, we can obtain the dynamic scoring result for the current moment. The details of each component are discussed as follows.

### 3.1 Dual Channel Text Encoding

Earlier research has highlighted the necessity to differentiate between users and products to create distinct representations (Ji et al., 2020). We consider that the global information of the text corresponds to the user, while the local information corresponds to the product. Hence, we adopt a

| Notations | Descriptions |
|---|---|
| $\mathcal{E}_i$ | The i-th review |
| $\mathbf{E}_u$ | Database for users |
| $\mathbf{E}_p$ | Database for products |
| $H$ | Text embedding of the review document |
| $H_u$ | User-related context |
| $H_p$ | Product-related context |
| $\mathbf{u}(t)$ | User's embedding at time t |
| $\mathbf{p}(t)$ | Product's embedding at time t |
| $\mathbf{u}(t^-)$ | User's embedding before time t |
| $\mathbf{p}(t^-)$ | Product's embedding before time t |
| $\mathbf{u}(t^- + \Delta_u)$ | Projected user embedding |
| $\mathbf{p}(t^- + \Delta_p)$ | Projected product embedding |
| $\tilde{\mathbf{p}}(t^- + \Delta_u)$ | Predicted user's next product embedding |

Table 1: Important notations and descriptions

parallel multi-scale representation learning method called MUSE (Zhao et al., 2019) to encode text into global and local channels for users and products respectively. We first embed the document $d = [x_1, x_2, \cdots, x_L]$ through BERT (Devlin et al., 2019) to get its pretrained word embeddings:

$$[\mathbf{h}_1, \mathbf{h}_2, \cdots, \mathbf{h}_L] = \text{BERT}(x_1, x_2, \cdots, x_L) \quad (1)$$

where $h_i$ is a d-dimensional feature vector for representing corresponding tokens $x_i$. We denote text representation as $H$. Then we follow the subsequent steps to obtain global and local context.

*Channel 1: Self-Attention Mechanism for User's Global Context.* We first project the document representation $H$ into into three parts, key $K$, query $Q$, and value $V$. The global context for users is then obtained by means of a self-attention mechanism:

$$H_u = \text{softmax}\left(\frac{Q_u K_u^T}{\sqrt{d_k}}\right) V_u \quad (2)$$

where $Q_u, K_u, V_u$ are transformed from $H$ through the linear layer.

*Channel 2: Convolution for Product's Local Context.* DynamicConv (Wu et al., 2019) is applied here, whose each convolution sub-module contains multiple cells with different kernel sizes to capture different-range features. The output of the convolution cell with kernel size k is:

$$\text{Conv}_k(H) = \text{Depth\_conv}_k(V_p) W^{\text{out}} \quad (3)$$

where $V_p$ is transformed from $H$ by a shared projection with *Channel 1*, and $\text{Depth\_conv}_k(V_p)$ is a traditional depthwise convolutional function, which

formulas as follows:

$$\text{Depth\_conv}_k(X) = \sum_{j=1}^{k} \left(\text{softmax}\left(\sum_{c=1}^{d} W_{j,c}^Q\right)\right. \quad (4)$$
$$\left. X_{i,c} \cdot X_{i+j-\lceil \frac{k+1}{2} \rceil, c}\right)$$

where the output corresponds to the calculation results of the document's $i$-th element of output channel $c$. In the end, the weight of different convolution cells is automatically selected through a gating mechanism for the product's local context:

$$H_p = \sum_{i=1}^{n} \frac{\exp(\alpha_i)}{\sum_{j=1}^{n} \exp(\alpha_j)} \text{Conv}_{k_i}(H) \quad (5)$$

### 3.2 Dual Channel Dynamic Graph Updating

Our proposed dual-channel dynamic graph updating component is primarily inspired by JODIE (Kumar et al., 2019), with adaptive adjustments and designs specially made for sentiment analysis on streaming user reviews.

Given a set of reviews with N users and M products, we first create two embedding lookup tables for both users and products as $\mathbf{E}_u = [\mathbf{u}_1, \ldots, \mathbf{u}_N]$ and $\mathbf{E}_p = [\mathbf{p}_1, \ldots, \mathbf{p}_M]$, which also act as a current representation storage database. At time 0, each user $u$ and product $p$ in the database is initialized randomly from a uniform distribution into an r-dimensional vector $\mathbf{u}(0) \in \mathbb{R}^r$ and $\mathbf{p}(0) \in \mathbb{R}^r$. When new input arrives, the representation stored in the database is taken out as $\mathbf{u}(t^-)$ and $\mathbf{p}(t^-)$ and added to the updating process. After this, the updated information will be renewed to the database as $\mathbf{u}(t)$ and $\mathbf{p}(t)$. Note that we only maintain the most recent representation of the users and products in these two databases.

The entire updating process is designed as a dual-channel structure of user-part and product-part, where the user part contains a projection module and an updating module, and the product part only has an updating module.

**User Part.** The first part for the user is a Projection Module to process temporal projections. We assume that user preference is continuously shifting even when there are no review actions. Therefore, for user embeddings after the elapsed time $\Delta_u$, we perform the following projection function:

$$\mathbf{u}(t^- + \Delta_u) = (1 + \mathbf{w}) * \mathbf{u}(t^-) \quad (6)$$

where $\mathbf{w} \in \mathbb{R}^r$ is a time-context vector converted from $\Delta_u$, and the larger $\Delta_u$ is, the more the projected embedding vector differs from the input embedding vector.

Later, a Next Product Prediction (NIP) cell, which predicts the product that the user is likely to review, is proposed to enhance the representation of the user:

$$\tilde{\mathbf{p}}(t^- + \Delta_u) = W_1 \mathbf{u}(t^- + \Delta_u) \\ + W_2 \mathbf{p}^-(t^-) + B \quad (7)$$

where $\mathbf{p}^-(t^-)$ is product embedding before time $t$ corresponding to the product from $u$'s previous review, and $W_1 \in \mathbb{R}^{r \times r}$, $W_2 \in \mathbb{R}^{r \times r}$ and $B \in \mathbb{R}^r$ are trainable parameters in linear layer.

For the NIP unit, our aim is to minimize the difference between the predicted product embedding $\tilde{\mathbf{p}}(t^- + \Delta_u)$ and the real product embedding $\mathbf{p}(t^- + \Delta_p)$, where $\Delta_p$ is the time difference between the current review and the last review for product $p$, which is different from $\Delta_u$. Since we assume that the products do not change during the time interval, here $\mathbf{p}(t^- + \Delta_p)$ equals $\mathbf{p}(t^-)$. Thus, the loss function can be represented as:

$$L_{nip} = \sum_{(u,p,t,d) \in \mathcal{E}} \left\| \tilde{\mathbf{p}}(t^- + \Delta_u) - \mathbf{p}(t^-) \right\|_2 \quad (8)$$

where we use the L2 loss function to push the predicted product representation closer to the true product representation.

The second part for the user is an Update Module, implemented based on an RNN cell, which generates the updated representation of the user after this review. It takes the projected user embedding $\mathbf{u}(t^- + \Delta_u)$, the previous product embedding $\mathbf{p}(t^-)$, and the user's time interval $\Delta_u$ as input, then integrates them into the current user representation:

$$\mathbf{u}(t) = \tanh\big(W_1^u\big(\mathbf{p}(t^-) \oplus \Delta_u \oplus H_u\big) \\ + W_2^u \mathbf{u}(t^- + \Delta_u) + b_u\big) \quad (9)$$

where $W_1^u \in \mathbb{R}^{(r+1+d) \times r}$, $W_2^u \in \mathbb{R}^{r \times r}$ and $b_u \in \mathbb{R}^r$ are parameters of user's update RNN cell. Through these two modules, the user's representation flows smoothly from $t^-$ to $t$.

**Product Part.** The product part is similar to the user side, except that there is no projection module, as we consider the product to be static for the duration. In contrast to the user part, the positions of $\mathbf{u}(t^- + \Delta_u)$ and $\mathbf{p}(t^-)$ are switched, and the context used for product updating is $H_p$:

$$\mathbf{p}(t) = \tanh\big(W_1^p\big(\mathbf{u}(t^- + \Delta_u) \oplus \Delta_p \\ \oplus H_p\big) + W_2^p \mathbf{p}(t^-) + b_p\big) \quad (10)$$

where $W_1^p \in \mathbb{R}^{(r+1+d) \times r}$, $W_2^p \in \mathbb{R}^{r \times r}$ and $b_p \in \mathbb{R}^r$ are parameters of product's update RNN cell.

### 3.3 Training

In order to ensure the quality of user and product embeddings, we apply the L2 loss between $t^-$ and $t$ for users and products to prevent sudden changes in continuous time:

$$L_{u_{smooth}} = \left\| \mathbf{u}(t) - \mathbf{u}(t^-) \right\|_2 \quad (11)$$

$$L_{p_{smooth}} = \left\| \mathbf{p}(t) - \mathbf{p}(t^-) \right\|_2 \quad (12)$$

where $\mathbf{u}(t)$ and $\mathbf{u}(t^-)$ represent the user's previous and current representation respectively, while $p$ corresponds to the product.

For score prediction at time $t$, we calculate cross-entropy loss between the predicted score $\widehat{y}$ and true score $y$:

$$L_{score} = -\frac{1}{N} \sum_{i=1}^{N} \sum_{c=1}^{K} y \log(\widehat{y}) \quad (13)$$
$$\text{where } \widehat{y} = \text{MLP}\big(\mathbf{u}(t) \oplus \mathbf{p}(t) \oplus H\big)$$

Finally, our training objective can be formulated as follows:

$$L = \lambda_1 L_{score} + \lambda_2 L_{nip} \\ + \lambda_3 L_{u_{smooth}} + \lambda_4 L_{p_{smooth}} \quad (14)$$

where $\lambda_1$, $\lambda_2$ and $\lambda_3$ as well as $\lambda_4$ are tradeoff parameters for each loss.

## 4 Experiments

We construct 5 datasets for sentiment analysis on streaming user reviews to evaluate the performance of our model and promote the development of this research. We first collect 5-core data from the Amazon Reviews, which are built by McAuley et al. (2015) and Ni et al. (2019) in a variety of categories to ensure the diversity of our datasets. The former dataset we refer covers the period from May 1996 to July 2014, and the latter ranges from May 1996 to Oct 2018. The statistics of the 5 datasets are shown in Table 2. For more construction details, see the Appendix.

We adopt BERT (base-uncased) (Devlin et al., 2019) as the pretrained encoder for texts. For user and product embeddings, we explore it in dimensions [8, 16, 32, 64, 128, 256]. The batch size is 8, and the learning rate is 3e-5, the number of epochs is 2. $\lambda_1$, $\lambda_2$, $\lambda_3$ and $\lambda_4$ are all set to 1. The dropout

| Catagory | Total Num | User Num | Product Num | Avg r/u | Avg r/p | Test Avg r/u | Test Avg r/p |
|---|---|---|---|---|---|---|---|
| *Books* | 25164 | 500 | 497 | 50.328 | 50.632 | 52.019 | 56.066 |
| *Grocery_and_Gourmet_Food* | 9725 | 500 | 500 | 19.45 | 19.45 | 20.456 | 24.088 |
| *Kindle_Store* | 11742 | 500 | 488 | 23.484 | 24.061 | 24.632 | 28.373 |
| *Sports_and_Outdoors* | 6002 | 500 | 493 | 12.004 | 12.174 | 12.278 | 15.537 |
| *reviews_CDs_and_Vinyl* | 19583 | 500 | 500 | 39.166 | 39.166 | 44.819 | 41.610 |

Table 2: Statistical information of our self-constructed datasets. 'Avg r/u' means the average associated reviews number for users, and 'Avg r/p' means the average associated reviews number for products. Test corresponding to the test set.

of MUSE is set to 0.1, and three different kernel size of [1, 3, 5] is applied for deepwise convolution. For more on our modifications to the JODIE structure, see the Appendix. We formalize the score prediction task as a classification problem, and the following seven evaluation indicators are used to evaluate our model: Accuracy, Precision, Recall, F1, MSE, RMSE, and MAE. For the split of the training set and test set, we adopt a ratio of 9:1.

## 4.1 Baseline Methods

As the task of sentiment analysis on streaming user reviews has not yet been investigated, we create two different baselines to contrast with our DC-DGNN method:

- **Text-based model**: The **Bert-Sequence** model, a Bert Model transformer (Devlin et al., 2019) with a sequence classification head on top, which is a frequent winner in classification tasks, is still applied here as a very competitive baseline. Similarly, the structure of the combination of Bi-directional LSTM with Attention (**BiLSTM+Att**) is also used as another baseline.

- **User and Product-based model**: **JODIE** (Kumar et al., 2019) applies a coupled recurrent neural network model to learn embedding trajectories of users and items, which has the ability to predict a state change in users of the recommendation system. Here we use it as a baseline for score prediction. Due to its t-batch strategy, only the regression task can be implemented here. **CHIM** (Amplayo, 2019) adopts a chunk-wise matrix representation for user/product attributes and injects user/product information in different locations. **IUPC** (Lyu et al., 2020) uses a dual channel modeling approach similar to ours by stacking text information into user and product through multi-head attention. **NGSAM** (Zhou et al.,

2021b) exploits group information to supplement user information by a group-wise model, which is designed as a regression task with bias.

## 4.2 Main Results

The experimental results on the datasets from May 1996 to October 2018 are shown in the first four rows of Table 3, and the dataset from May 1996 to July 2014 is shown in the last row of Table 3. It can be observed that:

1) On all five datasets, the performance of the Bert-based model (Bert-Sequence, IUPC, and our DC-DGNN) is better than that of the Glove-based model (BiLSTM+Att, NGSAM, and CHIM), which proves that a high-quality feature extractor is still a necessity for sentiment analysis.

2) Our DC-DGNN model outperforms all other baselines on 5 datasets, confirming the superiority of DC-DGNN in modeling user and product temporal dynamics. Meanwhile, compared to JODIE, our DC-DGNN has significant improvements, which shows that we have successfully adapted the dynamic graph structure to our sentiment analysis task.

3) Comparing the results on the first four datasets with the last dataset, we observe that our model shows convincible and considerable performance when datasets have a relatively longer timespan, which indicates that our DC-DGNN has the potential to model on more real-world ever-increasing streaming datasets.

## 4.3 Ablation Study

In order to verify the impact of the modules proposed in this paper, ablation experiments were designed as follows: 1) w/o DCTE means removing the Dual Channel Text Encoding (DCTE) structure and directly updating the user and product with

| Dataset | Method | Criteria | | | | | | |
|---|---|---|---|---|---|---|---|---|
| | | Accuracy(↑) | Precision(↑) | Recall(↑) | F1(↑) | MSE(↓) | RMSE(↓) | MAE(↓) |
| Books | BiLSTM+Att | 0.5793 | 0.4219 | 0.3528 | 0.3458 | 0.6726 | 0.8201 | 0.4970 |
| | Bert-Sequence | 0.6687 | 0.6032 | 0.4941 | 0.5269 | 0.4203 | 0.6483 | 0.3584 |
| | JODIE | - | - | - | - | 17.6734 | 4.2040 | 4.2035 |
| | NGSAM | - | - | - | - | 0.7027 | 0.8383 | 0.6401 |
| | CHIM | 0.6293 | 0.5338 | 0.4310 | 0.4557 | 0.6293 | 0.7933 | 0.4466 |
| | IUPC | 0.6722 | **0.6237** | 0.5130 | 0.5473 | 0.4168 | 0.6456 | 0.3548 |
| | **DC-DGNN** | **0.6814** | 0.6086 | **0.5363** | **0.5627** | **0.4044** | **0.6360** | **0.3449** |
| Grocery_and_Gourmet_Food | BiLSTM+Att | 0.6814 | 0.3925 | 0.3755 | 0.3816 | 0.8243 | 0.9079 | 0.4543 |
| | Bert-Sequence | 0.7235 | 0.5672 | 0.4032 | 0.4202 | 0.5036 | 0.7096 | 0.3453 |
| | JODIE | - | - | - | - | 19.5945 | 4.4266 | 4.4246 |
| | NGSAM | - | - | - | - | 0.6628 | 0.8141 | 0.5462 |
| | CHIM | 0.7050 | 0.5425 | 0.4514 | 0.4660 | 0.7328 | 0.8560 | 0.4080 |
| | IUPC | 0.7235 | 0.3879 | 0.3666 | 0.3563 | 0.5971 | 0.7727 | 0.3669 |
| | **DC-DGNN** | **0.7451** | **0.6207** | **0.5191** | **0.5021** | **0.4892** | **0.6994** | **0.3227** |
| Kindle_Store | BiLSTM+Att | 0.5804 | 0.3325 | 0.3561 | 0.3337 | 0.7864 | 0.8868 | 0.5226 |
| | Bert-Sequence | 0.6511 | 0.5540 | 0.4897 | 0.5086 | 0.4230 | 0.6504 | 0.3719 |
| | JODIE | - | - | - | - | 17.4456 | 4.1768 | 4.1761 |
| | NGSAM | - | - | - | - | 0.5454 | 0.7385 | 0.5515 |
| | CHIM | 0.6485 | 0.4535 | 0.3986 | 0.4035 | 0.6026 | 0.7762 | 0.4238 |
| | IUPC | 0.6545 | 0.5672 | 0.4994 | 0.5198 | 0.4383 | 0.6620 | 0.3719 |
| | **DC-DGNN** | **0.6664** | **0.5912** | **0.5157** | **0.5370** | **0.3991** | **0.6318** | **0.3532** |
| Sports_and_Outdoors | BiLSTM+Att | 0.6689 | 0.1843 | 0.2016 | 0.1637 | 1.2396 | 1.1134 | 0.5441 |
| | Bert-Sequence | 0.6872 | 0.3879 | 0.3447 | 0.3545 | 0.5391 | 0.7342 | 0.3794 |
| | JODIE | - | - | - | - | 19.8610 | 4.4566 | 4.4558 |
| | NGSAM | - | - | - | - | 0.7356 | 0.8577 | 0.5840 |
| | CHIM | 0.6905 | 0.4392 | 0.3109 | 0.3341 | 0.9967 | 0.9983 | 0.4742 |
| | IUPC | 0.6872 | 0.4034 | 0.3713 | 0.3791 | 0.5541 | 0.7444 | 0.3844 |
| | **DC-DGNN** | **0.7038** | **0.4227** | **0.3950** | **0.4063** | **0.5025** | **0.7089** | **0.3561** |
| reviews_CDs_and_Vinyl | BiLSTM+Att | 0.6605 | 0.4227 | 0.3988 | 0.3939 | 0.8474 | 0.9205 | 0.4655 |
| | Bert-Sequence | 0.7182 | 0.5635 | 0.5218 | 0.5396 | 0.4436 | 0.6660 | 0.3252 |
| | JODIE | - | - | - | - | 18.5957 | 4.3123 | 4.3091 |
| | NGSAM | - | - | - | - | 0.7320 | 0.8556 | 0.6885 |
| | CHIM | 0.7009 | 0.5340 | 0.4803 | 0.4912 | 0.5365 | 0.7325 | 0.3640 |
| | IUPC | 0.7070 | 0.5363 | 0.5056 | 0.5177 | 0.4273 | 0.6537 | 0.3313 |
| | **DC-DGNN** | **0.7243** | **0.5691** | **0.5435** | **0.5542** | **0.4084** | **0.6390** | **0.3134** |

Table 3: Main experimental results of our DC-DGNN model and comparison with previous works on datasets in the May 1996 - Oct 2018 time period (first 4 datasets) and May 1996 - July 2014 time period (last dataset). ↓ indicates the smaller the metrics, the better the method, while ↑ indicates the larger the metrics, the better the method. The score marked as bold means the best performance among all the methods.

| | Books | | Grocery_and_Gourmet_Food | | Kindle_Store | | Sports_and_Outdoors | | reviews_CDs_and_Vinyl | |
|---|---|---|---|---|---|---|---|---|---|---|
| | RMSE(↓) | MAE(↓) | RMSE(↓) | MAE(↓) | RMSE(↓) | MAE(↓) | RMSE(↓) | MAE(↓) | RMSE(↓) | MAE(↓) |
| w/o DCTE | 0.6553 | 0.3643 | **0.6648** | **0.2980** | 0.6451 | 0.3566 | 0.7664 | 0.3977 | 0.6406 | 0.3175 |
| w/o user&product | 0.6553 | 0.3651 | 0.7096 | 0.3124 | 0.6588 | 0.3745 | 0.7477 | 0.3993 | 0.6446 | 0.3267 |
| DC-DGNN | **0.6360** | **0.3449** | 0.6994 | 0.3227 | **0.6318** | **0.3532** | **0.7089** | **0.3561** | 0.6390 | **0.3134** |

Table 4: Compare the results under the original structure with the results of w/o DCTE and w/o user&product.

the BERT cls representation; 2) w/o user&product removes user and product information, and only completes score prediction through dynamically learned text representation. As shown in Table 4, w/o DCTE and w/o user&product all cause a performance drop on each dataset except for *Grocery_and_Gourmet_Food*, where its text content is relatively short, and MUSE in DCTE generally has advantages in encoding long texts rather than short texts. We also find that both w/o DCTE and user&product result in significant drops on *Sports_and_Outdoors*, indicating that even with relatively sparse user/product information but a long timespan, DC-DGNN's dual modeling of user and product dynamics still plays a crucial role in score prediction.

## 4.4 Variants of DC-DGNN Structure

**Discussion of User and Product Embedding Dimensions.** Different datasets have their own char-

The tale centers on Buttercup, a beautiful young farm girl and Wesley, a farm hand who fall in love with each other. But in the country of Florin there reposes evil in the form of Prince Humpernick and his aide Count Rugen. Many people will have seen the movie as well and thus know the basic plot. Still they will find delight in reading the book.

Anyone interested in this genre will most likely enjoy his books, several of which have been made into films. I am not a particularly big fan of such books, but have read a few, including several of his earlier ones. Four stars interesting and entertaining, but not deeply meaningful. I recommend that you read his novels in the order in which they were written and continue for as long as you find them interesting.

I decided to read Paula Hawkins The Girl on the Train because it received quite good critical reviews. But after about 50 pages I found it to be depressing and difficult to follow since the story keeps switching among various people and dates. I skipped to the end and read the last 25 pages just to see how it ended. Rachel is divorced from Tom and is both depressed and an alcoholic. As the plot unfolds the characters interact with each other in negative ways to the tragic ending.

★☆☆☆☆

| | |
|---|---|
| CHIM | 5 |
| BERT | 3 |
| IUPC | 3 |
| JODIE | 4 |
| **OURS** | 1 |

09 26, 2016  ★★★★★     01 27, 2017  ★★☆☆☆     02 10, 2017  ?

Figure 3: The case study (grey indicates neutral expression, red indicates positive expression, and green indicates negative expression).

acteristics in distribution and domain, so it is unreasonable to set the same embedding dimension for modeling on all datasets. To this end, we conduct a dimension exploration experiment to compare the impact of different dimension sizes on 5 datasets. Overall in Figure 4, we can see that the better results (i.e., the darker the blue color) are basically on the right side of the graph, indicating that in most cases, it's appropriate to set a relatively large dimension. This is because our dataset is basically at the level of ten thousand, with relatively abundant data, and modeling with large dimensions is beneficial. However, the largest is not always the best when some datasets are relatively small. For example, *Sports_and_Outdoors* achieves the best results when the dimension is 64.

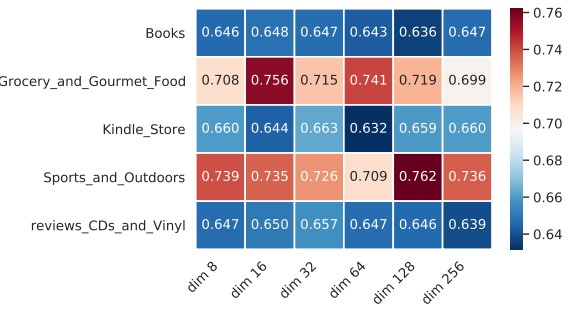

Figure 4: The visualize RMSE scores by setting the user and product dimensions to [8, 16, 32, 64, 128, 256] on different datasets

**Variants of Processing Text Information for DCTE.** After obtaining global and local contexts through DCTE, it has an additional dimension related to sequence length compared to user and product representation. To align them, we have multiple processing methods. Other than the *cls* result reported in the main experiment, we also tried two different methods: *mean* and *max*. *cls* takes the

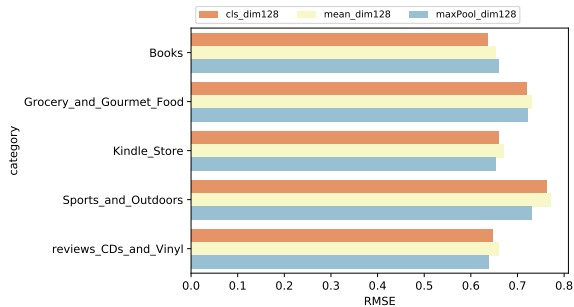

Figure 5: RMSE comparison of different text processing methods in DCTE

encoding at position 0 of the global $H_u$ and local representations $H_p$ respectively for updating users and products; *mean* averages the representation of the entire dimension 1 of $H_u$ and $H_p$; *max* conducts max_pool1d on the entire dimension 1. For the sake of fairness, we compare their performance with both the user and product dimensions set to 128. Figure 5 shows that, in general, *cls* can achieve relatively good results. However, in specific, the method corresponding to the best results still depends on the characteristics of different datasets.

### 4.5 Case Study

Figure 3 shows a user's reviews sampled from the *Books* dataset. The goal is to predict the rating score from the same user of the review on February 10, 2017. CHIM makes the worst prediction that differs the most from the actual 1-star. This is because CHIM focuses on attribute incorporation and cannot obtain user- or product-biased sentiment information. IUPC and BERT get the same results, indicating that IUPC cannot satisfy the need for user dynamics modeling when the user characteristic is complex. JODIE's result also indicates that its original structure is not an appropriate setting,

as explained in detail in the Appendix. Our model, on the contrary, can accurately capture the user's constantly negative scoring trend when the three expressions are very close and predict a score of 1, which shows the importance of dynamic modeling.

## 4.6 Computational Efficiency

In this section, we will discuss the time efficiency comparison of our proposed DC-DGNN and other models. We conduct a time test to compare our model's performance with that of IUPC and CHIM, which also learn user and product embeddings simultaneously. As shown in Table 5, DC-DGNN demonstrates better computational efficiency on most datasets. This can be attributed to the continuous storage of updated embeddings in a database, in contrast to previous methods that train two separate embedding instances. As a result, our method's time complexity is $O(L \cdot e) + O(L \cdot H \cdot d)$, which is a reduction compared to the $O(D \cdot L \cdot H \cdot d)$ complexity of other methods. Furthermore, the time advantage becomes even more obvious as any of these dimensions increases. While CHIM, which relies on GloVe (Pennington et al., 2014) encoding, may show faster processing on specific datasets, it requires a larger number of training epochs.

| Category | DC-DGNN | IUPC | CHIM |
|---|---|---|---|
| *Books* | **9.3172** | 12.1095 | 21.5937 |
| *Grocery_and_Gourmet_Food* | **3.3225** | 3.7920 | 2.2807 |
| *Kindle_Store* | **4.1851** | 4.2269 | 4.7659 |
| *Sports_and_Outdoors* | 2.0984 | 2.1240 | **1.2126** |
| *reviews_CDs_and_Vinyl* | **7.1875** | 7.7507 | 14.4805 |

Table 5: Comparison of Training Time (minutes/epoch)

## 5 Conclusion

In this paper, we present novel research on sentiment analysis on streaming user reviews and propose a dual-channel dynamic graph neural network, namely DC-DGNN, to model the temporal dynamics of users and products. DC-DGNN dynamically updates user and product representations through the dual-channel evolution of the graph. On our 5 self-constructed datasets, by comprehensive evaluations and ablation study, we confirm the superiority of our DC-DGNN and the impact of its modules. Through additional analytical experiments, we further demonstrate the importance of modeling user and product dynamics, hence verifying the conjecture in this paper.

## Limitations

Although our model has shown excellent performance in sentiment analysis on streaming user reviews, we still believe it has some limitations:

- Comparing datasets with longer timespan and shorter timespan, we find that improvement is not noticeable for the datasets with shorter timespan, which is a limitation for analysis only with short-term data. At the same time, our DC-DGNN model is also not friendly for datasets with sparse user/product information.

- In early experimental attempts, for the structure built directly from JODIE, we found significant differences in prediction performance at different times. When making predictions on data with a relatively recent time, the performance is great, and when the time is farther away, the performance is sharply decreased.

- The current structure we proposed considers forward integration and ignores backpropagation. In fact, the addition of subsequent reviews will also have an impact on past node representation. At the same time, we also ignore the high-order correlations between user-user and product-product. For example, user and user can be connected through a second-order homogeneous graph. The above-mentioned more refined design in graph updating may be our future improvement direction.

## Acknowledgement

The authors would like to thank the anonymous reviewers for their insightful comments. This work is funded by the National Natural Science Foundation of China (62176053). This work is supported by the Big Data Computing Center of Southeast University.

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

## A Experiment Details

**Dataset construct details.** Firstly, the data was subjected to a cleaning operation where we removed duplicates, as well as the data where their texts are empty, and then removed special characters from the texts. Later, the processing steps to build the datasets for sentiment analysis on streaming user reviews are as follows:

- Step1: Sort users by the number of reviews they have made, then filter the data containing the top 500 users from the original data according to their review counts.

- Step2: For the data obtained in the first step, sort the products according to the number of reviews they received, and filter the data containing the top 500 products.

- Step3: Normalise the data. For time processing, convert it to timestamp format and subtract the smallest timestamp in the dataset. For users and products, map the unique identifier to a numeric number.

- Step4: Finally, the data are sorted by timestamp to fit in our chronological setting.

**The reason why *Text-based model* is necessary here.** Although many previous methods that incorporate user and product information have shown strong potential, however, in some cases, they not only fail to learn high-quality representations but have counterproductive effects on score prediction. For example, in some datasets, only a few products are widely reviewed, and the number of reviews received by different products shows an imbalance. In fact, this data imbalance is very common in the real world, and even if we try to avoid it when constructing datasets, it still exists to some extent. Therefore, in this paper, we still take *text-based model* as a strong baseline.

**Experimental adjustments for adapting JODIE to sentiment analysis tasks.** The original JODIE structure first divides the data by timespan and then further divides the data by t-batch. T-batch is a strategy to prevent the overlapping of users and items in each batch. However, we believe this strategy may be inappropriate as it disrupts the original time order, which contradicts the chronological setup. To address this, we only divide the data into multiple time periods according to timespan and abandon the t-batch strategy to ensure the original order of the data.

Additionally, JODIE has a problem with gradients vanishing during testing. As JODIE was designed for recommendation systems, it keeps gradients propagation during testing, which is not ideal for our sentiment analysis task. To solve this, we propose a strategy that saves and updates the user and product representations in the databases during training. And during the testing phase, we only take the representations out of the database without updating them.

In JODIE, there is also a small experimental problem. We find that its static embedding setting conflicts with successful training. JODIE sets one-hot vectors as static embedding for each user and item in a rude way, which is not friendly to the situation where there are a large number of users and items, and will directly cause calculation overload and make it impossible to train. Therefore, in this paper, we deal with this problem by directly abandoning static embedding.

## B Supplement Experimental Results

Based on the unified dynamic graph library implemented by Yu et al. (2023), we present in Figure 6 a comparative analysis of our DC-DGNN alongside several other dynamic graph models. **CAWN** (Wang et al., 2021) leverages causal anonymous walks and recurrent neural networks to extract node representations by exploring network dynamics and causality. **DyGFormer** (Yu et al., 2023) utilizes a neighbor co-occurrence encoding scheme and a patching technique to capture long-term temporal dependencies. **GraphMixer** (Cong et al., 2023) incorporates a fixed-time encoding function into a link encoder, and employs a node encoder with neighbor mean-pooling for feature summarization. From the results, it is obvious that their performance is actually on par with JODIE's, which is unsatisfactory. All of these models also rely on text information for training, similar to our model. However, it is clear that they do not fully exploit the potential of review texts, resulting in subpar task performance, which further reflects the significance of our work.

| Dataset | Method | Criteria | | | | | | |
|---|---|---|---|---|---|---|---|---|
| | | Accuracy(↑) | Precision(↑) | Recall(↑) | F1(↑) | MSE(↓) | RMSE(↓) | MAE(↓) |
| *Books* | JODIE | 0.4849 | 0.1927 | 0.2046 | 0.1409 | 1.5655 | 1.2512 | 0.7977 |
| | CAWN | 0.3144 | 0.0629 | 0.2000 | 0.0957 | 0.9756 | 0.9877 | 0.7688 |
| | DyGFormer | 0.4785 | 0.0957 | 0.2000 | 0.1295 | 1.5965 | 1.2635 | 0.8112 |
| | GraphMixer | 0.0201 | 0.0040 | 0.2000 | 0.0079 | 11.1066 | 3.3327 | 3.1888 |
| | **DC-DGNN** | **0.6814** | **0.6086** | **0.5363** | **0.5627** | **0.4044** | **0.6360** | **0.3449** |
| *Grocery_and_ _Gourmet_Food* | jodie | 0.1942 | 0.0634 | 0.2116 | 0.0871 | 1.6637 | 1.2898 | 1.0323 |
| | CAWN | 0.2080 | 0.0416 | 0.2000 | 0.0689 | 1.0748 | 1.0367 | 0.8703 |
| | DyGFormer | 0.0302 | 0.0060 | 0.2000 | 0.0117 | 6.7165 | 2.5916 | 2.4585 |
| | GraphMixer | 0.0240 | 0.0048 | 0.2000 | 0.0094 | 12.5374 | 3.5408 | 3.4104 |
| | **DC-DGNN** | **0.7451** | **0.6207** | **0.5191** | **0.5021** | **0.4892** | **0.6994** | **0.3227** |
| *Kindle_Store* | JODIE | 0.3589 | 0.1394 | 0.2005 | 0.1111 | 1.0006 | 1.0003 | 0.7416 |
| | CAWN | 0.3606 | 0.0721 | 0.2000 | 0.1060 | 0.9216 | 0.9600 | 0.7195 |
| | DyGFormer | 0.0380 | 0.0076 | 0.2000 | 0.0147 | 5.5645 | 2.3589 | 2.2027 |
| | GraphMixer | 0.0210 | 0.0042 | 0.2000 | 0.0082 | 10.8859 | 3.2994 | 3.1607 |
| | **DC-DGNN** | **0.6664** | **0.5912** | **0.5157** | **0.5370** | **0.3991** | **0.6318** | **0.3532** |
| *Sports_and_ _Outdoors* | JODIE | 0.2427 | 0.1988 | 0.2065 | 0.0971 | 1.3662 | 1.1689 | 0.9224 |
| | CAWN | 0.2056 | 0.0411 | 0.2000 | 0.0682 | 1.0911 | 1.0446 | 0.8733 |
| | DyGFormer | 0.0189 | 0.0038 | 0.2000 | 0.0074 | 7.1000 | 2.6646 | 2.5622 |
| | GraphMixer | 0.0300 | 0.0060 | 0.2000 | 0.0117 | 13.1044 | 3.6200 | 3.5022 |
| | **DC-DGNN** | **0.7038** | **0.4227** | **0.3950** | **0.4063** | **0.5025** | **0.7089** | **0.3561** |
| *reviews_CDs_ _and_Vinyl* | JODIE | 0.1600 | 0.0402 | 0.1557 | 0.0639 | 2.2033 | 1.4844 | 1.2033 |
| | CAWN | 0.2491 | 0.0498 | 0.2000 | 0.0798 | 1.1668 | 1.0802 | 0.8652 |
| | DyGFormer | 0.5830 | 0.1166 | 0.2000 | 0.1473 | 1.5650 | 1.2510 | 0.6991 |
| | GraphMixer | 0.0364 | 0.0073 | 0.2000 | 0.0141 | 11.9721 | 3.4601 | 3.3009 |
| | **DC-DGNN** | **0.7243** | **0.5691** | **0.5435** | **0.5542** | **0.4084** | **0.6390** | **0.3134** |

Table 6: Supplement experimental results of our DC-DGNN model and comparison with previous dynamic graph works on datasets in the May 1996 - Oct 2018 time period (first 4 datasets) and May 1996 - July 2014 time period (last dataset). ↓ indicates the smaller the metrics, the better the method, while ↑ indicates the larger the metrics, the better the method. The score marked as bold means the best performance among all the methods.