# OpenReview forum: "Sentiment Analysis on Streaming User Reviews via Dual-Channel Dynamic Graph Neural Network"
_EMNLP/2023/Conference — EMNLP 2023 Main_

### Official Review · Reviewer_axgv · 2023-08-02

**Soundness:** 3

**Excitement:**

3: Ambivalent: It has merits (e.g., it reports state-of-the-art results, the idea is nice), but there are key weaknesses (e.g., it describes incremental work), and it can significantly benefit from another round of revision. However, I won't object to accepting it if my co-reviewers champion it.

**Paper Topic And Main Contributions:**

The paper proposes to consume time information when performing sentiment analysis of reviews as users' preferences and products' aspects are changing over time. The proposed model adopts temporal gnn to incorporate temporality of users and products representation vectors as reviews are coming from the past to the current time interval in order to predict the sentiment of a user toward a product in the future yet-to-be-observed time interval.

**Questions For The Authors:**

Please see "reason to reject".

**Reasons To Accept:**

+ Well-structured and readable paper
+ Sound formalization and evaluation methodology
+ Adequate review categories for the training datasets

**Reasons To Reject:**

- [resolved] ~~My main concern is the lack of a temporal baseline. All the paper's baselines are non-temporal. I understand that there might not be a direct temporal study in sentiment analysis, but temporality has been well-studied in temporal recommender systems. At least 2-3 of such baselines, of course, with adaptation for the task of the paper, should be included.~~

- The motivation for the temporal study is somehow not related to the current work. The authors claim that "the users’ preferences and the popularity of products are continuously changing over time"-L#055, which is true but not related. In review analysis, a user essentially writes a review about her experience of using the product. The popularity of a product or her preference may not have a direct influence on her review. At least this motivation needs more clarity by explaining it in Figure 1.

- [resolved if included in paper] ~~The experiment lacks information about the settings of the temporal study. While it includes hyperparameters of the model, it is missing what were the time intervals (yearly, I assume), which part was used for training (I assume streams of reviews before up until time t), the test part (I assume the t+1), etc ...~~

- No promise to release the codebase has been made for the reproducibility of the work.

**Reproducibility:**

3: Could reproduce the results with some difficulty. The settings of parameters are underspecified or subjectively determined; the training/evaluation data are not widely available.

**Reviewer Confidence:**

3: Pretty sure, but there's a chance I missed something. Although I have a good feel for this area in general, I did not carefully check the paper's details, e.g., the math, experimental design, or novelty.

**Typos Grammar Style And Presentation Improvements:**

-- not sure I understood "self-build" dataset. They were all Amazon dataset of different product categories. Do you mean you order them in time or what?
-- not sure I understood "act as a current representation storage database"?
-- I would suggest use "r" for review and "d" for dimension :)
-- there are few typos in terms of punctuations and the likes

---

> ### Author Rebuttal · Authors · 2023-08-29
>
> We thank the reviewer for the comments. The major concerns of the reviewer are answered below.
>
> > My main concern is the lack of a temporal baseline. All the paper's baselines are non-temporal. I understand that there might not be a direct temporal study in sentiment analysis, but temporality has been well-studied in temporal recommender systems. At least 2-3 of such baselines, of course, with adaptation for the task of the paper, should be included.
>
> The JODIE reported in the paper belongs to one of the baselines you mentioned. In addition, in the table below we supplement some baselines of Dynamic Graph Models in recommender systems adapted to this task. However, it is worth mentioning that their performance is actually in line with JODIE's performance, which is not good. In fact, these models also rely on text information to learn, but it is obvious that they do not really make full use of the text information and perform poorly on the task, which further reflects the significance of our work.
>
> - CAWN - Wang et al. ICLR2021
>
> - DyGFormer - Yu et al. arxiv2023
>
> - GraphMixer - Cong et al. ICLR2023
>
>
> | Category | Model | accuracy | precision | recall | f1  | MSE | RMSE | MAE |
> | --- | --- | --- | --- | --- | --- | --- | --- | --- |
> | Books | CAWN | 0.3144 | 0.0629 | 0.2 | 0.0957 | 0.9756 | 0.9877 | 0.7688 |
> |     | DyGFormer | 0.4785 | 0.0957 | 0.2 | 0.1295 | 1.5965 | 1.2635 | 0.8112 |
> |     | GraphMixer | 0.0201 | 0.004 | 0.2 | 0.0079 | 11.1066 | 3.3327 | 3.1888 |
> |     | **OURS** | **0.6814** |**0.6086**| **0.5363** | **0.5627** | **0.4044** | **0.6360** |**0.3449**|
> | Grocery_and_Gourmet_Food | CAWN | 0.208 | 0.0416 | 0.2 | 0.0689 | 1.0748 | 1.0367 | 0.8703 |
> |     | DyGFormer | 0.0302 | 0.006 | 0.2 | 0.0117 | 6.7165 | 2.5916 | 2.4585 |
> |     | GraphMixer | 0.024 | 0.0048 | 0.2 | 0.0094 | 12.5374 | 3.5408 | 3.4104 |
> |     | **OURS** | **0.7451** | **0.6207** | **0.5191** | **0.5021** | **0.4892** | **0.6994** | **0.3227** |
> | Kindle_Store | CAWN | 0.3606 | 0.0721 | 0.2 | 0.106 | 0.9216 | 0.96 | 0.7195 |
> |     | DyGFormer | 0.038 | 0.0076 | 0.2 | 0.0147 | 5.5645 | 2.3589 | 2.2027 |
> |     | GraphMixer | 0.021 | 0.0042 | 0.2 | 0.0082 | 10.8859 | 3.2994 | 3.1607 |
> |     | **OURS**| **0.6664** | **0.5912** | **0.5157** | **0.5370** | **0.3991** | **0.6318** | **0.3532** |
> | Sports_and_Outdoors | CAWN | 0.2056 | 0.0411 | 0.2 | 0.0682 | 1.0911 | 1.0446 | 0.8733 |
> |     | DyGFormer | 0.0189 | 0.0038 | 0.2 | 0.0074 | 7.1 | 2.6646 | 2.5622 |
> |     | GraphMixer | 0.03 | 0.006 | 0.2 | 0.0117 | 13.1044 | 3.62 | 3.5022 |
> |     | **OURS** | **0.7038** | **0.4227** | **0.3950** | **0.4063** | **0.5025** | **0.7089** | **0.3561** |
> | reviews_CDs_and_Vinyl | CAWN | 0.2491 | 0.0498 | 0.2 | 0.0798 | 1.1668 | 1.0802 | 0.8652 |
> |     | DyGFormer | 0.583 | 0.1166 | 0.2 | 0.1473 | 1.565 | 1.251 | 0.6991 |
> |     | GraphMixer | 0.0364 | 0.0073 | 0.2 | 0.0141 | 11.9721 | 3.4601 | 3.3009 |
> |     | **OURS**| **0.7243** | **0.5691** | **0.5435** | **0.5542** | **0.4084** | **0.6390** | **0.3134** |
>
> > The motivation for the temporal study is somehow not related to the current work. The authors claim that "the users’ preferences and the popularity of products are continuously changing over time"-L#055, which is true but not related. In review analysis, a user essentially writes a review about her experience of using the product. The popularity of a product or her preference may not have a direct influence on her review. At least this motivation needs more clarity by explaining it in Figure 1.
>
> As we illustrate in the example, what the proposed method captures is the overall change of the user over time, including the user's interest and the user's language habits, which is directly related to sentiment analysis.
>
> > The experiment lacks information about the settings of the temporal study. While it includes hyperparameters of the model, it is missing what were the time intervals (yearly, I assume), which part was used for training (I assume streams of reviews before up until time t), the test part (I assume the t+1), etc ...
>
> Following the division method of the previous dynamic graph work, we first convert the time at the daily level into a timestamp, and then divide the data into a training set and a test set of 9:1 according to the timestamp. This method is adopted to ensure that during the training process, the model reads a relatively stable stream of data to avoid situations where there is too much or too little data. Also, the review stream before $t$ is used as the training set, and $t+1$ is used as the test set, but here $t$ represents the timespan divided according to the equal proportion of the timestamp, not a specific year or day.
>
> > No promise to release the codebase has been made for the reproducibility of the work.
>
> We will release the code upon the acceptance of the paper.

---

### Official Review · Reviewer_tvcR · 2023-08-03

**Typos Grammar Style And Presentation Improvements:** 1. The meaning of p^{-}(t^{-}) in Lin…
**Soundness:** 3

**Excitement:**

2: Mediocre: This paper makes marginal contributions (vs non-contemporaneous work), so I would rather not see it in the conference.

**Paper Topic And Main Contributions:**

The main contribution of this paper falls into the "NLP engineering experiment" category. This paper aims to detect the sentiment of online streaming reviews. The authors claim that existing works neglect the time-varying nature of this scenario and propose DC-DGNN to capture user and product dynamics for sentiment analysis. DC-GNN designs a dual-channel dynamic graph updating module to model the incoming user-product review pairs and updates the representations of users and products accordingly. Experimental results demonstrate the effectiveness of the proposed model.

**Questions For The Authors:**

Q1: Is it necessary to build complex models to capture user and product dynamics to infer the sentiment of the given text? Does this task focus more on detection rather than prediction?

Q2: Why is only the regression performance reported in Table 3?

Q3: When dealing with online scenarios, can DC-DGNN deal with unseen users and products?

**Reasons To Accept:**

A1. Sentiment analysis of online streaming reviews has practical implications in various applications and is worth studying.

A2: The model adopts effective existing modules to capture user and product dynamics and yields promising performance in the experiment part.

A3: The authors compare their model with extensive baselines and provide an in-depth analysis of the proposed model.

**Reasons To Reject:**

R1: The motivation of this paper is not convincing. Existing works usually model the users and products to make predictions, while the task studied in this paper falls into the detection category. According to Line 182, the task aims to predict the user's rating y given the current condition that contains the review document. As I understand, the review document is almost sufficient to determine the sentiment. And the BERT performance also acknowledges the importance of review documents. I don't think it is essential to build complex models to capture user and product dynamics in such a scenario.

R2: The novelty of the model is limited. The main architecture comprises MUSE and JODIE with slight modifications.

R2.1: It is unclear how MUSE can capture global and local information for users and products respectively. No corresponding results are provided.

R2.2: The adaption of JODIE is questionable. In the original JODIE, the projection module is used for future prediction, while this paper takes it to update the representations of users and products. The design is unreasonable as I understand.

R3: Though extensive experiments are reported, the results cannot provide sufficient useful information to demonstrate the superiority of the proposed model except for Section 4.2. Incorporating more tasks or diving deeper into the model's performance of capturing user and product dynamics may help. An illustration of the user/product embedding's trajectory can also help.

**Reproducibility:**

3: Could reproduce the results with some difficulty. The settings of parameters are underspecified or subjectively determined; the training/evaluation data are not widely available.

**Reviewer Confidence:**

4: Quite sure. I tried to check the important points carefully. It's unlikely, though conceivable, that I missed something that should affect my ratings.

---

> ### Author Rebuttal · Authors · 2023-08-29
>
> We thank the reviewer for the comments. The major concerns of the reviewer are answered below.
>
> > R1: The motivation of this paper is not convincing. Existing works usually model the users and products to make predictions, while the task studied in this paper falls into the detection category. According to Line 182, the task aims to predict the user's rating y given the current condition that contains the review document. As I understand, the review document is almost sufficient to determine the sentiment. And the BERT performance also acknowledges the importance of review documents. I don't think it is essential to build complex models to capture user and product dynamics in such a scenario.
> >
> > Q1: Is it necessary to build complex models to capture user and product dynamics to infer the sentiment of the given text? Does this task focus more on detection rather than prediction?
>
> **On the one hand, we found through analysis that the user and product representation learned by the previous work on the insufficient use of text information is surface.** These representations may perform well on ranking prediction tasks that are less difficult, but they perform poorly on more difficult classification tasks. Therefore, building a sophisticated way to learn more refined user and product representations is actually our real purpose, and it can be used for both prediction tasks (when there is no given text at the next moment) and detection tasks (when there is a given text at the next moment).
>
> **On the other hand, sentiment analysis of streaming user reviews is a very valuable problem.** Previous work on sentiment analysis did not distinguish temporal tags. In practical scenarios, when dealing with user reviews in 2019, it is unrealistic to rely on reviews in 2020 for model training. In fact, experiments also prove that when we switch the setting from shuffle to temporal, the performance of many previously excellent models has dropped significantly. For example, as shown in the table (accuracy), the IUPC model performs very well in the shuffle setting, but drops significantly in the temporal setting, emphasizing the need to study the sentiment analysis under the temporal setting.
>
> |     | IUPC-shuffled | IUPC-temporal |
> | --- | --- | --- |
> | reviews_CDs_and_Vinyl_5_top500_sorted | **0.7167** | **0.707** |
> | Grocery_and_Gourmet_Food_5_top500_sorted | **0.7636** | **0.7235** |
> | Sports_and_Outdoors_5_top500_sorted | **0.7155** | **0.6872** |
>
> > R2: The novelty of the model is limited. The main architecture comprises MUSE and JODIE with slight modifications.
> >
> > R2.1: It is unclear how MUSE can capture global and local information for users and products respectively. No corresponding results are provided.
> >
> > R2.2: The adaption of JODIE is questionable. In the original JODIE, the projection module is used for future prediction, while this paper takes it to update the representations of users and products. The design is unreasonable as I understand.
>
> In fact, model transfer between different fields is not as simple as it seems. In this process, we spent a lot of time, from input to intermediate processing to final testing, all of which have undergone a lot of modifications. As is shown in the paper, the performance gap between the proposed method and JODIE is huge. During the experiments, we found that the gap between the recommendation system field and the NLP field is actually very large. **Eliminating this gap and applying the dynamic graph model to the streaming user review sentiment analysis is a big part of our work.** We are the first to effectively bridge the gap and successfully achieve the goal.
>
> - *Regarding the modification of JODIE*, in fact, we have also done a lot of analysis and changes. Although it may not be seen in the paper, in the actual code implementation, we almost replaced the entire implementation framework of JODIE, making it suitable for sentiment analysis in the NLP Task field. Throughout the work, we have conducted a detailed study of what the original paper describes and what the code implements. Since the user has no other information except the id tag, even if the word "prediction" is used for the projection module in the original JODIE, we tend to understand it as a "simulation of the future trajectory" rather than a real prediction. As a matter of fact, we have identified implementation errors in various other aspects of JODIE. For example, the training strategy mentioned in the appendix and numerous details regarding embedding updates. We have modified JODIE and improved it.
>
> - *For the usage of MUSE*, we use such description: "We consider that the global information of the text responds to the user, while the local information corresponds to the product.". The most common explanation of MUSE is that it has a self-attention that captures global features and a depth-separable convolution for capturing local features, whereas "the convolution concentrates on the full use of local information, while the self-attention focuses on capturing dependencies" is recognized. In practice, when applying MUSE, we assume that, for users, the emphasis lies on comprehending the entire text, whereas for products, the focus is on specific details. The utilization of MUSE is motivated by our intention to employ an unsupervised method to create a distinction between user and product representations, aiming to address the inherent issue of overly similar representations present in the original JODIE model.
>
>
> Hence, the proposed method is not a simple combination of JODIE and MUSE, but a further development based on them with carefully design.
>
> > R3: Though extensive experiments are reported, the results cannot provide sufficient useful information to demonstrate the superiority of the proposed model except for Section 4.2. Incorporating more tasks or diving deeper into the model's performance of capturing user and product dynamics may help. An illustration of the user/product embedding's trajectory can also help.
>
> We will include a demonstration figure of user/product embedding's trajectory in the paper:
>
> 1. We will divide the timeline into three timespans, where each timespan contains a portion of negative users (positive text but low ratings), positive users (negative text but high ratings), and neutral users (others).
> 2. We will extract user IDs for each of the three timespans using predefined rules, highlighting that these user IDs change across different timespans.
> 3. Utilizing T-SNE visualization, we will showcase the user embeddings learned by our DC-DGNN across these three timespans. Notably, the user IDs belonging to the same cluster across different timespans will correspond to the user IDs extracted in step 2.
>
> > Q2: Why is only the regression performance reported in Table 3?
>
> Since we use a lot of space to show all the evaluation indicators in Table 2, in order to control the length of the paper, we choose to show only regression scores in Table 3.
>
> > Q3: When dealing with online scenarios, can DC-DGNN deal with unseen users and products?
>
> Since we don't know the more detailed portrait information of users and products, the way we learn dynamic user and product embeddings is identified by a unique id, which means that it cannot handle unseen users and products. In fact, limited by the additional portrait information, many works do the same. If there are multiple portrait information, the method of combining multiple characteristics to form embedding can be used to deal with the unseen situation.

---

### Official Review · Reviewer_XFAW · 2023-08-04

**Soundness:** 4

**Excitement:**

4: Strong: This paper deepens the understanding of some phenomenon or lowers the barriers to an existing research direction.

**Missing References:**

A recent BERT-based model for rating prediction (sentiment analysis) should be included in the related work.

Qiu, Z., Wu, X., Gao, J., & Fan, W. (2021). U-BERT: Pre-training User Representations for Improved Recommendation. Proceedings of the AAAI Conference on Artificial Intelligence, 35(5), 4320-4327.

**Paper Topic And Main Contributions:**

This paper proposes a sentiment classifier for streaming user reviews by designing dynamic graph neural network. The experimental results show the superiority of the proposed model.

**Reasons To Accept:**

1. This work aims to address an interesting task, sentiment analysis on streaming user reviews, and proposes a suitable method to tackle the task.
2. The classification accuracy of the proposed model is the highest compared to the numbers of baseline models including recent BERT-based models.
3. The paper is well written so that it is easy-to-follow.

**Reasons To Reject:**

1. ~~The paper does not discuss on the computational cost of the proposed approach. The sentiment analysis on user reviews usually requires fast computation due to a large volume of reviews in real services such as Amazon and Netflix. It will be good to include computation complexity or running time comparison.~~

2. The experiments are highly biased to the heavy users and items. In the data preprocessing, this work only considers 500 heavy users and items that are associated with many reviews. Thus, the experiment shows the effectiveness of the proposed model only on the heavy users and items, while most users and items do not have many reviews.

3. ~~In the main experimental results (Table 2), the performance of JODIE is problematic. Its MAE is over 4, which is unusual with considering the Amazon datasets have 5-scale ratings. The paper has to discuss the issue to convince readers.~~

**Reproducibility:**

4: Could mostly reproduce the results, but there may be some variation because of sample variance or minor variations in their interpretation of the protocol or method.

**Reviewer Confidence:**

4: Quite sure. I tried to check the important points carefully. It's unlikely, though conceivable, that I missed something that should affect my ratings.

---

> ### Author Rebuttal · Authors · 2023-08-29
>
> We thank the reviewer for the comments. The major concerns of the reviewer are answered below.
>
> > The paper does not discuss on the computational cost of the proposed approach. The sentiment analysis on user reviews usually requires fast computation due to a large volume of reviews in real services such as Amazon and Netflix. It will be good to include computation complexity or running time comparison.
>
> The proposed approach is computational-efficient because  updated embeddings are continuously stored in a database, while previous methods rely on training two separate nn.Embedding. As a result, the time complexity of our method is O(L * e) + O(L * H * d), which is relatively smaller than O(D * L * H * d) of other methods. This time difference increases as any of these dimensions grows. We also conducted time tests, where only the IUPC and CHIM models concurrently learn user and product embeddings, serving as our points of comparison. As shown in the table below (measured in miniute/epoch), our model's performance is superior. While CHIM, which relies on GLove encoding, is faster on specific datasets, it requires more training epochs.
>
> |     | OURS | IUPC | CHIM |
> | --- | --- | --- | --- |
> | Books | 9.3172 | 12.1095 | 21.5937 |
> | Grocery_and_Gourmet_Food | 3.3225 | 3.7920 | 2.2807 |
> | Kindle_Store | 4.1851 | 4.2269 | 4.7659 |
> | Sports_and_Outdoors | 2.0984 | 2.1240 | 1.2126 |
> | reviews_CDs_and_Vinyl | 7.1875 | 7.7507 | 14.4805 |
>
> > The experiments are highly biased to the heavy users and items. In the data preprocessing, this work only considers 500 heavy users and items that are associated with many reviews. Thus, the experiment shows the effectiveness of the proposed model only on the heavy users and items, while most users and items do not have many reviews.
>
> Although we chose the top 500 users, according to the statistics of the datasets in Table 4 in the appendix, the avg r/u is not high on some datasets, such as only 12 on the Sports_and_Outdoors dataset, which is not considered as heavy users or items. Indeed, how to deal with low-resource users is a problem worthy of further exploration. We will study the model performance comparison between high-resource users and low-resource users later.
>
> > In the main experimental results (Table 2), the performance of JODIE is problematic. Its MAE is over 4, which is unusual with considering the Amazon datasets have 5-scale ratings. The paper has to discuss the issue to convince readers.
>
> The original JODIE structure is designed to learn user and product embeddings in recommender systems, and its task design and framework design are different from ours. We observe that the user and product representations learned by JODIE are very similar to an unusually close degree, and we tend to think that it only learns the surface representation. The JODIE results we reported in the paper are based on the original implementation version of JODIE (without earlystop), and the latest JODIE (with earlystop) is based on the revised version of Yu et al. 2023. Updated results of JODIE are not as bad as we ran out (as follows table), but not good either.
>
> |     | accuracy | precision | recall | f1  | MSE | RMSE | MAE |
> | --- | --- | --- | --- | --- | --- | --- | --- |
> | Books | 0.4849 | 0.1927 | 0.2046 | 0.1409 | 1.5655 | 1.2512 | 0.7977 |
> | Grocery_and_Gourmet_Food | 0.1942 | 0.0634 | 0.2116 | 0.0871 | 1.6637 | 1.2898 | 1.0323 |
> | Kindle_Store | 0.3589 | 0.1394 | 0.2005 | 0.1111 | 1.0006 | 1.0003 | 0.7416 |
> | Sports_and_Outdoors | 0.2427 | 0.1988 | 0.2065 | 0.0971 | 1.3662 | 1.1689 | 0.9224 |
> | reviews_CDs_and_Vinyl | 0.1600 | 0.0402 | 0.1557 | 0.0639 | 2.2033 | 1.4844 | 1.2033 |

---

### Meta-Review · Area_Chair_pXrH · 2023-09-17

**Recommendation:** 4

**Metareview:**

The reviewers are in general agreement that this paper could be accepted in EMNLP-2023. The reviewers also raised some issues with its presentation and motivation. The authors have provided a rebuttal that appeared to alleviate some concerns. A suggestion was made to accept the paper as the main conference or findings. The reviewers' concerns are also suggested to be addressed in the final version.

---

### Decision · Program_Chairs · 2023-10-07

**Decision:**

Accept-Main

**Comment:**

The reviewers are in general agreement that this paper could be accepted in EMNLP-2023. The reviewers also raised some issues with its presentation and motivation. The authors have provided a rebuttal that appeared to alleviate some concerns. A suggestion was made to accept the paper as the main conference or findings. The reviewers' concerns are also suggested to be addressed in the final version.